# Dyskerin Downregulation Can Induce ER Stress and Promote Autophagy via AKT-mTOR Signaling Deregulation

**DOI:** 10.3390/biomedicines10051092

**Published:** 2022-05-08

**Authors:** Daniela Maiello, Marianna Varone, Rosario Vicidomini, Valentina Belli, Marina De Rosa, Paola Dama, Maria Furia, Mimmo Turano

**Affiliations:** 1Department of Biology, University of Naples Federico II, 80126 Naples, Italy; daniela.maiello@unina.it (D.M.); marianna.varone@unina.it (M.V.); mfuria@unina.it (M.F.); 2Section on Cellular Communication, Eunice Kennedy Shriver National Institute of Child Health and Human Development, National Institutes of Health, Bethesda, MD 20892, USA; 3Medical Oncology, Department of Precision Medicine, Università degli Studi della Campania “Luigi Vanvitelli”, 80131 Naples, Italy; valentina.belli@hotmail.com; 4Department of Molecular Medicine and Medical Biotechnology, University of Naples Federico II, 80131 Naples, Italy; marina.derosa@unina.it; 5School of Life Sciences, University of Sussex, Brighton BN1 9QG, UK; p.dama@sussex.ac.uk

**Keywords:** dyskeratosis congenita, DKC1, autophagic flux, UPR, LC3 puncta

## Abstract

Dyskerin is an evolutionarily conserved nucleolar protein implicated in a wide range of fundamental biological roles, including telomere maintenance and ribosome biogenesis. Germline mutations of *DKC1*, the human gene encoding dyskerin, cause the hereditary disorders known as X-linked dyskeratosis congenita (X-DC). Moreover, dyskerin is upregulated in several cancers. Due to the pleiotropic functions of dyskerin, the X-DC clinical features overlap with those of both telomeropathies and ribosomopathies. In this paper, we evaluate the telomerase-independent effects of dyskerin depletion on cellular physiology by using inducible DCK1 knockdown. This system allows the downregulation of *DKC1* expression within a short timeframe. We report that, in these cellular systems, dyskerin depletion induces the accumulation of unfolded/misfolded proteins in the endoplasmic reticulum, which in turn induces the activation of the PERK branch of the unfolded protein response. We also demonstrate that the PERK-eIF2a-ATF4-CHOP signaling pathway, activated by dyskerin downregulation, triggers a functional autophagic flux through the inhibition of the PI3K/AKT/mTOR pathway. By revealing a novel unpredicted connection between the loss of dyskerin, autophagy and UPR, our results establish a firm link between the lowering of dyskerin levels and the activation of the ER stress response, that plays a key role in the pathogenesis of several diseases.

## 1. Introduction

Dyskerin is the major evolutionarily conserved pseudouridine synthase involved in the conversion of specific uridines to pseudouridines on ribosomal RNAs (rRNAs), small nuclear RNAs (snRNAs) and messenger RNAs (mRNAs) [1]. The protein associates with the three highly conserved proteins GAR1, NHP2 and NOP10, and with small ncRNAs bearing the H/ACA motifs (mostly snoRNAs), to form diverse functional nuclear ribonucloprotein (RNPs) complexes, including active telomerase, H/ACA small nucleolar ribonucleoproteins (snoRNPs) and the small Cajal body-specific ribonucleoproteins (scaRNPs). These complexes are involved in several biological functions that include both the safeguarding of telomere integrity and ribosome biogenesis [2]. Dyskerin is also involved in a wide range of biological processes, including IRES-dependent translation, [3,4,5], DNA damage response [6,7,8], vesicular trafficking [9] and energy metabolism [10]. In addition to these functions, dyskerin has been found upregulated in several human sporadic cancers [11,12,13] and its overexpression has been linked to poor prognosis [14]. Considering the multiple biological functions in which the protein is involved, it is not surprising that hypomorphic mutations of DKC1, the human gene encoding dyskerin, causes the hereditary disorder X-linked dyskeratosis congenita (X-DC) [15]. The main symptoms of this pathology are the progressive failure of proliferating tissues, premature aging and increased susceptibility to various types of cancers [16,17]. Due to the pleiotropic functions of dyskerin [18], the X-DC clinical features overlap with those of both telomeropathies and ribosomopathies. For example, cells from X-DC patients display a telomere shortening, but also altered snoRNA regulation and rRNA modification [18]; accordingly, certain dyskerin pathogenic X-DC variants substantially reduce the accumulation of hTR, but, in some cases, also that of other H/ACA RNAs [19]. In transgenic mice carrying hypomorphic DKC1 mutations, impaired ribosomal RNA pseudouridylation was observed before the reduction in telomere length, suggesting that deregulated ribosome function is important in the initiation of X-DC [17]. In addition, in organisms without telomerase, in which telomeres are maintained by retrotransposons, mutations in the human DKC1 orthologue cause context-dependent defects, suggesting that the telomerase-independent functions of pseudouridine synthases are important both for development and tissue homeostasis [20,21].

The use of cellular systems that allow the detection of which physiological/metabolic alterations are triggered by dyskerin loss of function, before a significant shortening of the telomeres occurs, represents a useful tool to evaluate the telomerase-independent functions of dyskerin. The timely identification of early alterations could lead to the identification of new molecular targets and, possibly, new therapeutic approaches to mitigate the most severe symptoms of this disease. In this paper, to detect the early (telomerase-independent) effects caused by dyskerin reduction, we use an inducible cellular silencing system capable of depleting dyskerin in human colon carcinoma (RKO) [9] and human embryonic kidney 293 cells (HEK 293T). Since a previous study has shown that ribosomes purified from dyskerin-depleted cells display altered translational fidelity when added to a cell-free system [5], our study focused on testing, in cellular systems, whether the depletion of dyskerin causes an accumulation of misfolded/unfolded proteins with the consequent activation of the pathways responsible for the mitigation of such accumulation.

Importantly, consistent with the accumulation of misfolded/unfolded proteins, we observe that dyskerin depletion triggers endoplasmic reticulum (ER) stress with the consequent activation of the unfolding protein response (UPR), eIF2α phosphorylation and ATF4 production. In addition, by using a GFP-LC3-RFP reporter transgene coupled with our own custom R-pipeline designed to precisely analyze the ratio RFP/GFP signal of LC3 puncta, we also discovered that dyskerin depletion causes a strong upregulation of the autophagic flux, which correlates with a strong downregulation of the mTOR signaling.

Altogether, our data show that the early effects triggered by dyskerin silencing concern the alteration of three important cellular pathways, namely UPR, autophagy and mTOR signaling. This leads us to suggest that players involved in these pathways, such as PERK (the kinase that phosphorylates eIF2a), or mTORC1/mTORC2, could represent molecular targets to be evaluated for the early diagnosis of X-DC, before the manifestation of late severe symptoms, such as bone marrow failure, might occur.

## 2. Materials and Methods

### 2.1. Generation of Stable, Dox-inducible DKC1 Silenced Cell Lines

To generate stable cell lines, HEK 293T cells were seeded in a 60 mm dish one day before transfection in a complete growth medium to achieve a confluence of 60–80% at the time of transfection. On the following day, 3 µg of pLKO-Tet-On-shDKC1 plasmid [9] and 12 µL of Polyethylenimine (PEI) (Polysciences, Valley Rd, Warrington, PA, USA) were diluted in 0.5 mL of serum-free Dulbecco Modified Eagle Medium (DMEM), respectively. After 5 min of incubation at room temperature, the diluted DNA and the diluted PEI were mixed and incubated for 30 min at room temperature, in order to allow the formation of the DNA/PEI complexes. Cells were then incubated with 2 mL of growth medium without antibiotics and 1 mL of DNA/PEI complexes at 37 °C in a humidified atmosphere of 5% CO_2_ for 5 h. After incubation, the culture medium was replaced with a fresh culture medium containing 20% fetal bovine serum (FBS), 1% penicillin-streptomycin (P/S) and 1% L-glutamine. Following 48 h of transfection, cells were passaged 1:10 into complete medium containing 750 ng/mL puromycin (Sigma-Aldrich, St. Louis, MO, USA). After 20 days of puromycin selection, during which the culture medium was changed every 2 days, all transformed cell clones were pooled together and the inducible stable cell line (HEK-shDKC1) was expanded. Silencing was triggered by adding doxycycline (Dox, Sigma-Aldrich, St. Louis, MO, USA) (400 ng/mL) to the medium.

### 2.2. Cell Culture and Treatments

RKO and HEK 293T (ATCC, Manassas, VA, USA), RKO-shDKC1 and HEK-shDKC1 cell lines were maintained at 37 °C in a humidified atmosphere of 5% CO_2_ in complete DMEM containing 10% Tet-free fetal bovine serum (FBS), 1% penicillin-streptomycin and 1% L-Glutamine (Euroclone, Milano, Italy). In addition, the culture media of the RKO-shDKC1 and HEK-shDKC1 cell lines were supplemented with 500 ng/mL of puromycin (Sigma-Aldrich, St. Louis, MO, USA).

For 3BDO (3-Benzyl-5-((2-nitrophenoxy)methyl)-dihydrofuran-2(3H)-one) treatment, cells were washed in 1X DPBS and resuspended in complete Tet-free culture medium containing 400 ng/mL Dox. Following 72 h of incubation at 37 °C in a humidified atmosphere with 5% CO_2_, the culture medium was replaced with complete Tet-free culture medium containing 400 ng/mL Dox and 100 μM of 3BDO (Sigma-Aldrich, St. Louis, MO, USA) and cells were grown for a further 24 h.

### 2.3. Transfection with GFP-LC3-RFP Reporter Transgene

To detect the autophagic flux, both untreated and 48 h Dox–treated HEK-shDKC1 and RKO-shDKC1 cells, seeded on glass coverslips placed in 35 mm dishes, were transfected with 3 µg of GFP-mRFP-LC3 plasmid (GFP-LC3-RFP-Plasmid, Addgene, Arsenal Way, Watertown, USA, 21074, depositing lab Tamotsu Yoshimori) and 12 µL of PEI (Polysciences, Valley Rd, Warrington, PA, USA) as described above. Following 48 h of transfection, coverslips were rinsed in 1X DPBS, fixed with 4% PFA (paraformaldehyde) (Biochemica, Billingham, UK, A3813), mounted on glass slides with DAPI solution and then examined under a Zeiss confocal fluorescence microscope Zeiss LSM 700 (Zeiss, Oberkochen, Germany). Images of control and depleted cells were always acquired at the same settings.

### 2.4. Reverse Transcription Quantitative PCR (RT-qPCR)

Total RNA was extracted from RKO and HEK cells using the QIAzol Lysis Reagent (Qiagen, Hilden, Germany, 79306), according to the manufacturer’s guidelines. Reverse Transcription Quantitative PCR (RT-qPCR) was carried out as previously described [22] using the following primer pair sequences: DKC1 forward, 5′-GGCGGATGCGGAAGTAAT-3′, DKC1 reverse, 5′-CCACTGAGACGTGTCCAAC-3′; ATG12 forward, 5′-CTTTGCTCCTTCCCCAGACC-3′, ATG12 reverse, 5′-CACGCCTGAGACTTGCAGTA-3′; ATG5 forward, 5′-TGTCCTTCTGCTATTGATCCTGA-3′, ATG5 reverse, 5′-CCGGGTAGCTCAGATGTTCA-3′; HPRT1 forward, 5′-TGGTCAAGGTCGCAAGCTT-3′, HPRT1 reverse, 5′-AGTCAAGGGCATATCCTAC-3; HSP70 forward, 5′-TTTTTGTGGCTTCCTTCGTT-3′, HSP70 reverse, 5′-GAGTAGGTGGTGCCAAGATCA-3′; and HSP90 forward, 5′-GTTTGAGAACCTCTGCAAAAT-3′, HSP90 reverse 5′-CATGGAGATGTCACCAATCG-3′. Relative quantifications were normalized to the expression of HPRT1 endogenous control.

### 2.5. Flow Cytofluorimetric Analyses

HEK 293T-shDKC1 and RKO-shDKC1 cells untreated and treated with Dox for 96 h or with 60 mM LiCl for 24 h were trypsinized, counted, washed three times in DPBS 1X and fixed in ice-cold ethanol at −20 °C overnight. The cells were then washed twice with cold DPBS, resuspended in a hypotonic solution (0.1% Na-Citrate, 50 μg/mL RNAse, 50 μg/mL propidium iodide) and incubated for 30 min in the dark at RT. The DNA content of the labeled cells was measured using a BD Accuri C6 flow cytometer and data were analyzed using BD Accuri™ C6 Software.

### 2.6. Protein Analysis

For protein extraction, cells were pelleted by centrifugation at 2500 rpm for 5 min at 4 °C and resuspended in a RIPA lysis buffer (50.0 mM Tris pH 7.5, 150.0 mM NaCl, 5.0 mM EDTA, 1% NP40) supplemented with anti-phosphatase and anti-protease (cOmplete™, Mini, EDTA-free Protease Inhibitor Cocktail, Sigma-Aldrich, St. Louis, MO, USA). After 30 min of incubation in ice, cell lysates were centrifuged at 14.000 rpm for 15 min at 4 °C to eliminate insoluble material. The supernatants were transferred in clean tubes and stored at −80 °C. The protein concentration of extracts was determined by Bio-Rad Protein Assay (Dye Reagent Concentrate, Biorad, Hercules, CA, USA, following the manufacturer’s instructions. For Western blot analysis, samples were run on SDS-PAGE under reducing conditions, transferred onto 0.45 μm nitrocellulose filters (GE Healthcare, Little Chalfont, UK) and blocked with 5% BSA in Tris-buffered saline containing 0.1% Tween 20 (TBST). After blocking, blots were incubated for 2 h with the primary antibody diluted in TBST and then for 2 h with horseradish peroxidase-conjugated anti-rabbit/mouse/IgG. The following primary antibodies were used in this study: dyskerin (Elabscience Biotechnology Inc., Houston, TX, USA, E-AB-31251; 1:100), GAPDH (Abcam, CA, USA, ab56788; 1:100), GRP78/BiP (Thermo Fisher Scientific, Waltham, MA, USA, PA5-11418; 1:1000), phospho PERK (Elabscience Biotechnology Inc., Houston, TX, USA, E-AB-21296; 1:1000), phospho eIF2α (Cell Signaling Technology (CST), Heidelberg, Germany, 9721; 1:1000), ATF4 (Santa Cruz Biotechnology; 1:500), CHOP (DSHB—Developmental Studies Hybridoma Bank, University of Iowa, Iowa city, USA; CPTC-DDIT3-1-S; 1:500), HSP70 (Santa Cruz biotechnology, sc-25837; 1:1000), HSP90 (Stressgen Biotechnologies Corporation, Ann Arbor, MI, USA, SPA-846; 1:1000), LC3 (Abcam, CA, USA, ab51520; 1:2000), Beclin 1 (Elabscience Biotechnology Inc., Houston, TX, USA, E-AB-33743; 1:1000), PARP (Santa Cruz Biotechnology, Dallas, TX, USA, sc-7150; 1:1000), BAX (Cell Signaling Technology (CST), Heidelberg, Germany2772; 1:500), phospho 4E-BP1 (Cell Signaling Technology (CST), Heidelberg, Germany, 2855; 1:1000), phospho p70 S6K (Cell Signaling Technology (CST), Heidelberg, Germany9205; 1:1000), phospho AKT (Cell Signaling Technology (CST), Heidelberg, Germany, 9271; 1:1000), and phospho GSK3β (GeneTex, Inc. (North America), Alton Pkwy Irvine, CA, GTX-59576; 1:1000). The following secondary antibodies used in this study were: Anti-Mouse-HRP (Bethil Laboratories, Montgomery, TX, USA, A90-144P; 1:10,000) and Anti-Rabbit-HRP (Bethil Laboratories, Montgomery, TX, USA, A120-100P; 1:10,000). Bands were visualized by ECL (Biorad, Hercules, CA, USA, 1705061) using the ChemiDoc XRS+ System (Bio-Rad) and quantified with Image Lab Software (Bio-Rad).

### 2.7. Immunostaining and Image Capturing

Cells were seeded on glass coverslips placed in 6-well plates and treated for 96 h with Dox. They were then fixed with 3.7% paraformaldehyde for 10 min, permeabilized in 100% ice-cold methanol for 10 min at −20 °C and finally blocked with DPBS supplemented with 3% bovine serum albumin (BSA) for 30 min. After each step, cells were rinsed in DPBS and incubated with primary antibodies overnight at 4 °C, following which the cells were washed in DPBS and incubated with secondary antibodies at RT for 1 h on the next day. Glass coverslips were counterstained with DAPI, mounted on glass slides and examined under a Zeiss confocal fluorescence microscope (Zeiss LSM 700). The primary antibodies used were the following: Calreticulin (D3E6) XP (Cell Signaling Technology (CST), Heidelberg, Germany, 12238; 1:250), Dyskerin (Santa Cruz Biotechnology, Dallas, TX, USA, sc-373956; 1:1000), GRP78/BiP (Thermo Fisher Scientific, Waltham, MA, USA, PA5-11418; 1:500) and LC3 (Abcam, CA, USA, ab51520; 1:3000). The secondary antibodies used were: Cy3-AffiniPure Goat Anti-Rabbit IgG (H + L) (Invitrogen, Van Allen Way Carlsbad, CA, USA, A10520; 1:500) and Alexa Fluor 488 donkey anti mouse IgG (H + L) (Invitrogen, Van Allen Way Carlsbad, CA, USA, A21202; 1:1000).

### 2.8. Immunoreactivity Quantification

Dyskerin—To quantify dyskerin accumulation levels, confocal z-stack images of cells stained with DAPI and anti-dyskerin were converted to maximum projection images using ImageJ (Fiji distribution; https://imagej.net/software/fiji accessed on 30 March 2022). The DAPI and dyskerin channels were separated and the background was removed by applying a threshold and a mask specific for each channel. The nucleus volume of the each analyzed cell was selected by manually drawing a ROI surrounding the DAPI fluorescence. For each ROI/Nucleus, the RawIntDen in the dyskerin channel was measured. This value represents the total dyskerin immunoreactivity and is an estimation of the total amount of dyskerin contained in a nucleus (DAPI volume). The RawIntDen values were split by condition (Dox– and Dox+ 96 h), normalized and reported relative to the Dox–. The statistical analysis was performed using PRISM software applying a one-tailed *t*-test.

CRT and GRP78—ImageJ was used to convert the confocal z-stack images of cells stained with anti-CRT or anti-GRP78 in maximum projections. ROIs surrounding each of the captured cells were manually drawn. The background was removed by applying a threshold and a mask, both specific for the examined channel.

For each ROI/cell the RawIntDen in the CRT or GRP78 channel was measured. In order to have a measure of the amount of CRT or GRP78 not dependent on the cell size, the RawIntDen values were divided by the areas of the obtained mask (CRT or GRP78) inside each cell. The value RawIntDen/CRTArea (or RawIntDen/GRP78 Area), defined also as CRTdensity (or GRP78 density), is an estimation of the CRT accumulation levels in each cell. The statistical analysis of the normalized RawIntDen/CRTArea (or RawIntDen/GRP78Area) values, split by condition (Dox– and Dox+ 96 h), was performed using PRISM software applying a one-tailed *t*-test.

LC3—To measure LC3 levels, the RawIntDen/LC3 Area value for each cell was calculated as described above for CRT and GRP78. In addition, the values were split by condition (Dox– and Dox+), normalized and reported relative to the Dox– cells. Additionally, the find maxima function from ImageJ (Fiji version) was used to automatically detect the number of LC3 puncta per cell. Dividing the LC3 area in each cell by the number of LC3 puncta, a measurement of the size of LC3 puncta was obtained and, by dividing the RawIntDen/LC3 area by the number of LC3 puncta, a measurement of the amount of LC3 per puncta was obtained. The statistical analyses for all the calculated parameters were performed by using a one-tailed *t*-test using Prism software comparing the Dox– and the Dox+ 96 h conditions.

### 2.9. Evaluation of the Autophagic Flux Using the GFP-mRFP-Reporter

For quantifying the autophagic flux, for both conditions (Dox– and Dox+ 96 h), five of the imaged GFP-mRFP-LC3 transfected cells were manually isolated with ImageJ software (Fiji distribution; https://imagej.net/software/fiji accessed on 30 March 2022). For each of the isolated cells, the channels were separated and, for both of them (GFP and mRFP), the background was removed by subtracting the filtered median image (radius = 8 pixels) from the original image. After merging the two resulting channels, linear ROIs (of about two microns) were manually centered on each of the detected puncta in the RFP channel. The multiplot profile function (Fiji) was used to convert each cell’s punctum information into tables of pixel intensity changes along linear regions of interest (ROI) for each detected punctum. The procedure was applied once for RFP intensity, and once FOR GFP intensity for each punctum in each cell. Subsequently, all tables were imported into RStudio, and the area under RFP- and GFP-plot profile curves for each punctum was calculated using the function trapz from the pracma package. Indeed, in acidic lysosomes, the ratio GFPArea/RFPArea is strongly reduced due to the quenching of GFP. The GFP/RF area ratio for each punctum was then labelled according to the cell condition (Dox– or Dox+ 96 h), and all the labelled values were aggregated into a single table for each signal (Appendix A for RFP and GFP, respectively). Significant ratio differences between Dox– and Dox+ conditions were estimated using a one-tailed *t*-test in Prism software. Finally, the ggplot2 package plots all RFP simultaneously as well as GFP-intensity profiles and ROIs. The ggplot2 package was used to simultaneously plot all RFP- and GFP-intensity profiles along the ROIs. The reproducible procedure has been reproduced as R script (Appendix A: R code).

### 2.10. Statistical Analyses

Statistical analyses were performed using GraphPad Prism 6 software GraphPad Software, Inc. La Jolla, CA, USA). Differences among multiple groups were compared using a one-way ANOVA. Differences were considered statistically significant when *p* < 0.05. The level of statistical significance is indicated by asterisks * *p* < 0.05, ** *p* < 0.01 and *** *p* < 0.001. All experiments were performed in triplicate with the exception of the lithium chloride treatment, whose results were confirmed by two different methods (Western blot and flow cytometry).

## 3. Results

### 3.1. Dyskerin Downregulation Leads to the Accumulation of HSP90, HSP70 and Calreticulin Chaperones

A previous study has shown that under-pseudouridylated ribosomes purified from dyskerin-depleted human cells showed altered translational fidelity when added to a cell-free system [5]. However, the effects of this telomeric-independent impairment of translational fidelity on cellular physiology remain to be determined. Recently, to study the telomerase-independent functions of human dyskerin, we developed an inducible cellular silencing system (pLKO-Tet-On-shDKC1 silencing vector) capable of depleting dyskerin in human colon carcinoma (RKO) cells upon the addition of tetracycline or doxycycline in culture media [9]. In this paper, we applied this system to test the possibility that the reduced translational fidelity of under-pseudouridylated ribosomes may induce the accumulation of unfolded/misfolded proteins and consequently lead to ER stress. In order to uncover phenomena not restricted to a specific cell type, we also generated a HEK 293T-shDKC1 cell line in this study. To obtain this line, we transfected human embryonic kidney 293T cells (HEK 293T) with a pLKO-Tet-On-shDKC1 silencing vector [9]. Once stably transformed lines were obtained, to minimize clonal differences, all subsequent experiments were carried out on pools of transformed cell clones (see Materials and Methods).

To verify whether dyskerin depletion could trigger the production of incorrectly folded proteins in general, we evaluated both mRNA and protein levels of Hsp70 and Hsp90 after the induction of DKC1 silencing in RKO and HEK 293T cells. Hsp70 and Hsp90 are chaperones that are known to play central roles in the folding of newly synthesized proteins and in the refolding of misfolded or misaggregated proteins, thus orchestrating the so-called heat shock response [23,24,25]. Importantly, our RT-qPCR and Western blot analyses show that Hsp70 and Hsp90 are upregulated at both the transcriptional and translational levels in dyskerin-silenced cells (Figure 1A–D). Specifically, we observed that, in both RKO-shDKC1 and HEK 293T-shDKC1 cells, a reduction of ~70% of DKC1 mRNA, obtained after 96 h of induction, causes a significant increase in Hsp70 and Hsp90 mRNA accumulation levels (Figure 1A,C). Consistent with an increased folding capacity, these transcriptional changes are translated to increased Hsp70 and Hsp90 protein levels during a time course of 96 h (Figure 1B,D).

These data demonstrate that the activation of the heat shock response after dyskerin depletion is not restricted to one specific cell type. In addition, since our analyses refer to a time window of 96 h, a time too short for a significant shortening of telomeres to occur, the induction of heat shock response upon dyskerin silencing is not due to the telomerase-dependent dyskerin function and appears at the early onset of the X-DC pathology.

To test whether the observed increased folding capacity induced by DKC1 silencing is directly due to the accumulation of unfolded/misfolded proteins synthesized by under-pseudouridylated ribosomes, we verified if dyskerin depletion correlates with an increased immunoreactivity of calreticulin (CRT). CRT is an ER-localized Ca2+-binding chaperone that binds to misfolded glycoproteins to promote their refolding and is thus considered an endoplasmic reticulum stress (ERS) marker [26,27]. Our confocal analyses show a considerable upregulation of CRT in both RKO and HEK 293T dyskerin-silenced cells (Figure 1E,F). Specifically, in both systems, we found that, in comparison with uninduced controls, 96 h of dyskerin silencing caused an 84% reduction in dyskerin immunoreactivity and more than a 4-fold increase in CRT immunoreactivity.

Hence, it is reasonable to conclude that the lack of translational fidelity caused by dyskerin downregulation results in a telomerase-independent accumulation of unfolded proteins and in the consequent induction of chaperones involved in their refolding upon ERS.

### 3.2. Dyskerin Depletion Triggers Unfolding Protein Response

Since the accumulation of unfolded/misfolded proteins can activate the UPR pathway, with enormous consequences on the general metabolism of the cell, we evaluated, upon dyskerin depletion, the expression of the glucose-regulated protein 78 (GRP78 also referred as Bip or HSPA5), a master regulator of UPR pathways [28,29,30].

Consistent with the idea that dyskerin depletion causes the activation of UPR, our confocal analyses showed that a 60% reduction in dyskerin immunoreactivity (induced by 96 h of doxycycline administration to RKO cells) triggers a strong increase (more than 7-fold) in GRP78 immunoreactivity (Figure 2A). The GRP78 accumulation following dyskerin depletion in RKO was further confirmed by Western blot analysis (Figure 2B).

To confirm that dyskerin downregulation triggers UPR, we analyzed, by Western blot, the accumulation levels of the key markers of the PERK pathway pPERK, p-eIF2 α, ATF4 and CHOP, during a time course of 96 h of DKC1 silencing in RKO cells [31]. As expected, we found that dyskerin depletion results in increased levels of pPERK and p-eIF2a, concomitantly with the accumulation of ATF4 and CHOP. Importantly, we also found that a time course of 96 h of dyskerin depletion induces a similar UPR response in HEK cells (Figure 2C).

Altogether, these findings indicate that dyskerin depletion promotes UPR activation through the PERK pathway. Since this activation is not related to the shortening of telomeres, the physiological and metabolic changes caused by UPR activation could represent important steps occurring before the onset of X-DC typical features.

### 3.3. Dyskerin Depletion Does Not Induce Apoptosis

Severe and prolonged ERS conditions that prevent cells from restoring ER homeostasis can lead the UPR to activate the apoptotic program, leading to cell death [32,33]. ERS-induced apoptosis can occur via different pathways [34]. In order to evaluate if dyskerin depletion could trigger apoptosis, we examined the silenced cells directly by flow cytometry analysis. To set up the optimal conditions and discriminate apoptotic cells from live cells, we used RKO and HEK 293T untreated cells as internal controls in these experiments. When these cells were induced to undergo apoptosis by a 24 h treatment with 60 mM LiCl, a GSK3β inhibitor known to induce apoptosis in several human cell lines [35,36,37], they exhibited a high percentage of subdiploid cells (Figure 3A). In contrast, when Dox-treated RKO and HEK 293T cells were analyzed for their DNA content, they presented only a negligible percentage of the subdiploid peak (Figure 3A), indicating that they are not subjected to apoptosis. Next, we checked by Western blotting the cleavage of poly (ADP-ribose) polymerase-1 (PARP-1), a known apoptotic marker, and found it was not activated by dyskerin depletion in HEK 293T cells. In contrast, its induction was efficiently triggered by LiCl treatment (Figure 3B). Accordingly, no induction of the apoptotic marker Bax was observed upon dyskerin silencing, while it was sharply induced upon LiCl treatment (Figure 3B). Similar results were previously obtained by our group in dyskerin-depleted RKO cells [9].

Collectively, the above findings indicate that ERS caused by dyskerin depletion does not activate apoptosis, at least not after 96 h of DKC1 silencing.

### 3.4. Dyskerin Downregulation Promotes Autophagy

It is clear from a previous study that the accumulation of misfolded proteins can activate autophagy as a defensive mechanism for maintaining ER homeostasis [29]. We thus wondered whether the activation of the PERK pathway observed in the silenced cells could also promote autophagy. Therefore, we evaluated by Western blot the levels of Beclin-1 marker and, as shown in Figure 4A,C, found that its accumulation level dramatically increased more than 6-fold in both types of dyskerin-depleted cells after 96 h of silencing (92% of depletion efficiency).

We then evaluated the conversion of LC3-I to LC3-II that is commonly used as a marker for the activation of autophagy [38]. Intriguingly, we observed that, while in unsilenced cells LC3-II is undetectable, dyskerin depletion strongly increases LC3-II accumulation levels even after 48 h of silencing (83% depletion efficiency) (Figure 4A,C). However, during this time course, LC3-I levels are only slightly increased in both cell types, implying that dyskerin depletion causes autophagosome formation.

The conjugation between ATG12 and ATG5 is essential for LC3 lipidation to form LC3-II, and therefore for autophagosome formation [39]. We therefore evaluated the expression of ATG5 and ATG12 by qRT-PCR and found an increase in their expression in the silenced cells (Figure 4B,D).

To confirm that dyskerin reduction induces autophagosome formation, we evaluated, by confocal imaging LC3 immunoreactivity, the cellular area occupied by LC3 puncta, the number of LC3 puncta per cell (which represents the most commonly used assay for measuring the autophagosome number [40]), the puncta size and the intensity of LC3 in each detected punctum in both the control and silenced cells. Importantly, we observed that a ~60% reduction in nucleolar dyskerin in RKO cells leads to an about 3-fold increase in both the total levels of LC3 per cell and the number of LC3 puncta per cell (from ~30 to ~90, Figure 4E). This LC3 increment following dyskerin depletion was further confirmed by a similar increase in the LC3Area/CellArea (Appendix A). In addition, we observed a 30% increase in the amount of LC3 contained in each punctum (Appendix A) and a 1.5 fold increase in the puncta size (from 0.32 to 0.49 um2; Figure 4E).

Although we observed a similar response to dyskerin depletion in HEK cells, there were some differences with the results obtained with RKO cells. Specifically, we estimated that, in HEK cells, a 50% reduction in dyskerin caused a considerable increase (~17 fold) in the total levels of LC3 per cell, but only a ~2-fold increase in LC3 puncta per cell (Figure 4F). This indicates that each punctum contains a considerable amount of LC3 or that, in this cell type, our confocal resolution is not sufficient to separate LC3 puncta that are very close to each other. This hypothesis was confirmed by our estimation of LC3 puncta size (increased by ~3-fold, from ~0.12 to 0.32 um2; Figure 4F) and by the LC3 amount contained in each detected punctum or overlapped puncta (increased by more than 4-fold upon dyskerin reduction; Appendix A).

Altogether, our data clearly demonstrate that, as a general phenomenon, dyskerin depletion induces autophagosomes accumulation. This accumulation could be either due to an induction of autophagy or its blockage (i.e., inefficient fusion or decreased autophagosome degradation) [41]. To rule out the possibility that the observed increase in LC3-II could derive from an accumulation of autophagosomes due to the block of autophagosome–lysosome fusion, we transiently transfected control and silenced cells with the mRFP-GFP-LC3 tandem construct. This construct permits monitoring the different stages of autophagic flux, thereby allowing one to discriminate autophagosomes from autolysosomes [40,42]. When the tandem-tagged fluorescent proteins accumulated in these cells, both green and red fluorescent signals were detectable at the early stages of the autophagic process; however, at later stages, the GFP signal was quenched inside the acidic autolysosomes, while the RFP signal was still retained. Following silencing, a downward shift in the distribution of GFPArea/RFPArea values (Figure 5A,B) in both RKO-shDKC1 and HEK 293T-shDKC1 cells was clearly observed.

Taken together, these results unequivocally indicate that dyskerin depletion induces a functional autophagic flux, which is characterized by an efficient mature autophagolysosome formation.

In this paper, we propose a hybrid image-processing/statistical analysis pipeline for autophagic flux evaluation based on the quantification of the GFPArea/RFPArea. Coupling the pipeline with the experimental design presented in this paper allowed the successful evaluation of autophagic flux in silenced cells and highlighted the suitability of the ratio as an autophagy proxy.

### 3.5. Dyskerin Downregulation Promotes Autophagy through the Inhibition of AKT/mTOR Signaling

The mechanistic target of rapamycin (mTOR) is the master regulator of autophagy [43], and recent studies have shown that ERS induces autophagy through the suppression of this pathway [44,45,46,47]. Thus, in order to explore the possibility that dyskerin depletion could induce autophagy by mTOR inhibition, we investigated the protein expression of its downstream substrates, i.e., 4E-BP1 and P70S6K, by Western blot analysis. As shown in Figure 6A,B, the expression of both these mTOR downstream targets was markedly reduced in both RKO and HEK silenced cells, indicating that dyskerin depletion effectively inhibited TORC1.

Considering that ERS and UPR are interlaced with the mTOR pathway, and that a large number of studies demonstrated that ERS downregulates AKT activity [45], we hypothesized that, upon dyskerin downregulation, the activation of the ERS/UPR pathways inhibited AKT/mTOR, which in turn could be responsible for the activation of autophagy. To confirm this hypothesis, we analyzed by Western blot the expression of pAKT in the dyskerin-silenced cells and found a significant decrease in the expression of this protein compared to their untreated controls (Figure 6A,B). To further assess the role played by pAKT in this context, we treated the silenced cells with the MTOR activator 3-benzyl-5-((2-nitrophenoxy) methyl)–dihydrofuran-2(3H)-one (3BDO). Accordingly, treating the silenced cells with 3BDO rescued the expression of p-AKT, p-4EBP1 and p-P70S6K and also prevented the accumulation of LC3 II (Figure 6C).

These findings fully support the conclusion that the inhibition of the AKT/mTOR signaling pathway plays a key role in triggering the autophagic process observed in the dyskerin-silenced cells.

## 4. Discussion

In this study, by using stably transformed RKO and HEK 293T cell lines able to trigger DKC1 silencing in inducible manner, we detected early cellular alterations due to dyskerin depletion.

Since it is reasonable to consider that the telomere length is not significantly reduced during 96 h of induction, it is likely that the alterations we identified do not depend on the role that dyskerin plays as a component of the telomerase complex.

Importantly, we found that both cell lines used in this study respond early to dyskerin depletion by increasing their protein folding capacity, as evidenced by the observed concomitant strong upregulation of HSP70, HSP90 and calreticulin chaperones. The induction of these chaperones, which are present in abundance within the ER [48], suggested that the loss of function of dyskerin leads to an accumulation of misfolded/unfolded proteins. Consistently with this idea, we found that the GRP78 chaperone, a main marker of UPR activation [49], rapidly accumulated upon DKC1 silencing, demonstrating that the lowering of dyskerin levels can stimulate immediately both the ERS and UPR homeostatic responses. These results fit well with previous data indicating that ribosomes purified from dyskerin-depleted cells were intrinsically altered in translation [5], thus being able to promote the synthesis of proteins bearing altered reading frames, whose presence is known to affect proper folding [50]. Accordingly, it can be presumed that the activation of the ERS and UPR pathways is likely to be due to the reduced translation fidelity caused by dyskerin depletion, with the consequent accumulation of misfolded/unfolded proteins in the ER lumen.

In parallel with the increased accumulation of GRP78, we found that several key effectors of the PERK-dependent UPR pathway are upregulated upon dyskerin depletion. Importantly, in both RKO-shDKC1 and HEK 293T-shDKC1 cells, reduced levels of dyskerin were able to induce eiF2α phosphorylation and to increase ATF4 levels. PERK belongs to the eIF2α kinase subfamily and represents one of the major ER transmembrane stress-responsive proteins [51]. Upon disturbances in protein folding, PERK is activated by a trans-autophosphorylation [52,53,54]. Activated PERK phosphorylates eIF2α [55,56,57] with the consequent attenuation of global protein synthesis [58,59] and, on the contrary, the enhanced translation of only selective sets of mRNAs, including ATF4 [5]. In turn, ATF4 induces the multifunctional C/EBP-homologous protein (CHOP) transcription factor [60].

It is worth noting that the reduced ribosome pseudouridylation (caused by either DKC1 pathologic mutations or reduced dyskerin levels) has to date been reported to differently modulate IRES-dependent translation [3,4,61]; in addition, ribosomes derived from dyskerin-depleted cells, when tested in an in vitro system, proved to significantly affect only IRES-mediated translation. Nonetheless, the increased level of eIF2α phosphorylation we observed in the dyskerin-silenced cells suggests that the lowering of dyskerin levels could impact the rate of cap-dependent translation in cellular systems. Indeed, it is possible that the results obtained in an in vitro system do not fully reproduce the real context of the growing cells, where potential response mechanisms may be activated dynamically.

To the best of our knowledge, this is the first time that dyskerin depletion is reported to correlate with an increase in PERK and eIF2α phosphorylation. The overactivation of PERK-eIF2α phosphorylation, which occurs in several models of human diseases [51], led to the prediction that the pharmacological modulation of the PERK-dependent UPR signaling pathways may contribute to ameliorate the symptoms of X-DC. Importantly, several pieces of evidence have demonstrated that eIF2α phosphorylation is a central event for the stimulation of autophagy [62]. For example, it has been established that the induction of the autophagy flux by different compounds is accompanied by eIF2α phosphorylation, and that the blockage of this phosphorylation, by using a non-phosphorylatable mutant of eIF2α, strongly abolishes the autophagy. Conversely, the inhibition of eIF2α phosphatase is sufficient to stimulate autophagy [62]. Altogether, these data demonstrate that eIF2α phosphorylation is sufficient to induce autophagy, even if the molecular details of this induction remain unknown. Based on the relationship between eIF2α phosphorylation and autophagy, it is reasonable to believe that the autophagic process we observed upon dyskerin depletion relies on p-eIF2α and it is downstream of the PERK-dependent UPR pathway.

Several studies reported that the main function of UPR signaling pathways is to protect cells from unfavorable conditions caused by pathological changes. Moreover, UPR protects the organism by eliminating cells that were exposed to extreme ER stress via apoptosis. Indeed, once ERS cannot be resolved, the persistence of UPR is known to activate the apoptotic program [63]. Since during a time course of 96 h we did not detect any apoptosis in the DKC1 silenced cells, it is possible to conclude that the activation of the PERK-dependent UPR pathway upon DKC1 silencing has an adaptive and not a pro-apoptotic role. Accordingly, the induction of apoptosis was never observed by our and other groups in several different human cancer cell lines subjected to different dyskerin depletion systems [9,12,64,65].

Consistent with the adaptive role of the PERK-dependent UPR pathway activated following dyskerin depletion, we also found that dyskerin downregulation induces a functional autophagic flux, which is characterized by an efficient mature autophagolysosome formation. To follow the efficiency of the process, we carefully evaluated autophagosome–lysosome fusion by coupling imaging and R data analysis. Autophagy acts as a pro-survival mechanism to remove misfolded proteins and restore ER homeostasis.

Importantly, the UPR-induced deactivation of mTOR contributes to the downregulation of AKT/TSC/mTOR pathway and promotes ERS-induced autophagy [45,66]. Similarly, the inhibition of AKT and mTORC1 by several agents, including the monoclonal antibody cetuximab, can induce autophagy [67,68,69,70,71,72,73,74,75]. Furthermore, the PERK–CHOP mediated induction of the AKT inhibitor TRB3 may also contribute to the inhibition of AKT upon severe ERS [76,77].

Consistent with the idea that a chronic activation of UPR induces autophagy by inhibiting mTORC1, we found that the accumulation level of 4E-BP1 and p70S6K is markedly reduced in both RKO-shDKC1 and HEK 293T-shDKC1 silenced cells. Intriguingly, in our experiments the treatment of the silenced cells with the mTOR activator 3BDO was able to rescue not only the expression of the autophagy marker LC3 II, but also that of p-AKT and of the two mTOR targets p-eIF4E-BP1 and p-p70S6K. This finding indicates that the activation of autophagy in the dyskerin-depleted cells is mediated by the inhibition of the AKT/mTOR pathway.

This study, for the first time, revealed an interdependence between mTOR signaling and UPR during a pathological program triggered by reduced levels of dyskerin. It is worth noting that the interplay between mTOR and chronic UPR has been associated with various pathologies [71,78,79,80,81], justifying the preclinical evaluations of combined therapies [66]. For example, preliminary studies suggest that treatment with ER-stress-alleviating “chemical chaperones”, such as 4-phenylbutyric acid (PBA), combined with a low dosage of the mTORC1 inhibitor (everolimus or rapamycin) could be used for the treatment of nephropathies and for TSC patients who are not suitable for surgical intervention [80,81,82].

The results we present in this paper link, for the first time, the loss of dyskerin with autophagy, UPR and mTOR signaling. In addition to further strengthening the role played by this multifunctional protein in the cellular homeostasis and in the stress response [10], these data suggest that a combined use of drugs aimed at relieving/suppressing the UPR (such as PBA and the PERK antagonist GSK2606414) and inhibiting the mTOR pathway (rapamycin or everolimus) might prevent or delay some severe manifestations of X-DC. These approaches might be useful especially in cases in which the onset of severe X-DC symptoms occur late, largely beyond childhood [83,84].

## Figures and Tables

**Figure 1 biomedicines-10-01092-f001:**
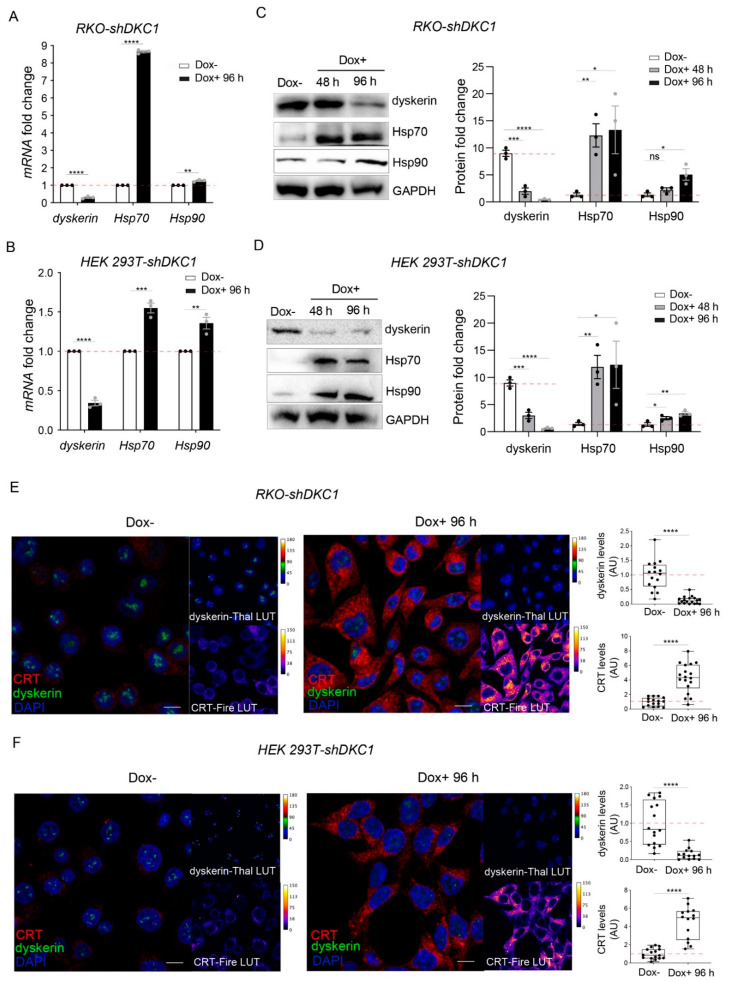
Dyskerin silencing stimulates the expression of chaperones involved in protein refolding. (**A**,**B**) qRT-PCR analysis showing the accumulation of Hsp70 and Hsp90 mRNAs in RKO-shDKC1 (**A**) and HEK 293T-shDKC1 cells (**B**) after DKC1 silencing induced by 96 h of Dox treatment. Data shown are relative to the untreated (Dox–) condition and as mean ± SEM (*n* = 3). (**C**,**D**) Western blot analysis showing an increased accumulation of Hsp70 and Hsp90 in RKO-shDKC1 (**C**) and HEK 293T-shDKC1 cells (**D**) after dyskerin depletion induced by 48 or 96 h of Dox treatment. The intensity of the bands was normalized to GAPDH and the differential accumulations evaluated with respect to the Dox– condition. Data are presented as mean ± SEM (*n* = 3). (**E**,**F**) Confocal images showing the increase in calreticulin (CRT) in RKO-shDKC1 (**E**) and HEK 293T-shDKC1 cells (**F**) treated with Dox for 96 h (Dox+) compared to their untreated control cells (Dox–). Scale bars 10 μm. Nuclei were counterstained with DAPI. CRT immunoreactivity is in red or Fire-Lut (false color Fire-Lookup Table) in separate channels. Dyskerin immunoreactivity is in green or in Thal-Lut (false color Thal-Lookup Table). CRT and dyskerin accumulation levels were quantified in multiple cells (*n* = 16 RKO-shDKC1 Dox–; *n* = 18 RKO-shDKC1 Dox+ 96 h; *n* = 16 HEK 293T-shDKC1 Dox–; *n* = 14 293T HEK 293T-shDKC1 Dox+ 96 h), normalized and reported relative to the Dox– condition as mean ± SEM. An unpaired one-tailed *t*-test was used in pairwise comparisons with respect to the Dox– condition in all statistical analyses in this figure (**** *p* < 0.0001; *** *p* < 0.001; ** *p* < 0.01; * *p* < 0.05; ns, *p* > 0.05).

**Figure 2 biomedicines-10-01092-f002:**
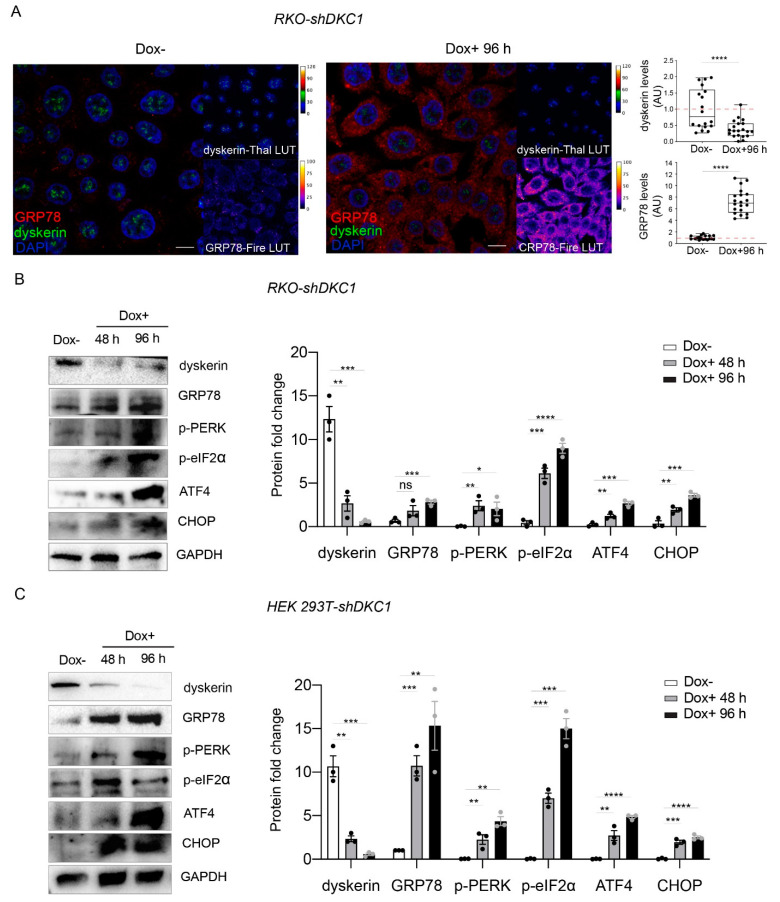
Dyskerin depletion induces ERS and increases UPR markers. (**A**) Confocal images showing the dramatic increase in GRP78/BiP (red or Fire-Lut) expression in RKO-shDKC1 cells treated with Dox for 96 h (Dox+, right panel) compared to untreated control cells (Dox–, left panel). Dyskerin (green or Thal-Lut) downregulation is evident from the strong reduction in the relative signal in Dox+ cells. Nuclei were counterstained with DAPI. GRP78/BiP and dyskerin levels were quantified in multiple cells (*n* = 18 RKO-shDKC1 Dox–; *n* = 20 RKO-shDKC1 Dox+ 96 h), normalized and reported relative to the Dox– condition as mean ± SEM. Scale bars 10 μm. (**B**,**C**) Western blot analysis confirming the upregulation of GRP78/BiP and showing the induction of key components of the PERK branch of UPR in RKO-shDKC1 (**B**) and HEK 293T-shDKC1 (**C**) Dox–treated (Dox+) cells for 48 or 96 h. Protein levels were normalized to GAPDH. Data are shown as mean ± SEM (*n* = 3). All statistical analyses were performed using an unpaired one-tailed *t*-test in pairwise comparisons with respect to the Dox– condition (**** *p* < 0.0001; *** *p* < 0.001; ** *p* < 0.01; * *p* < 0.05; ns, *p* > 0.05).

**Figure 3 biomedicines-10-01092-f003:**
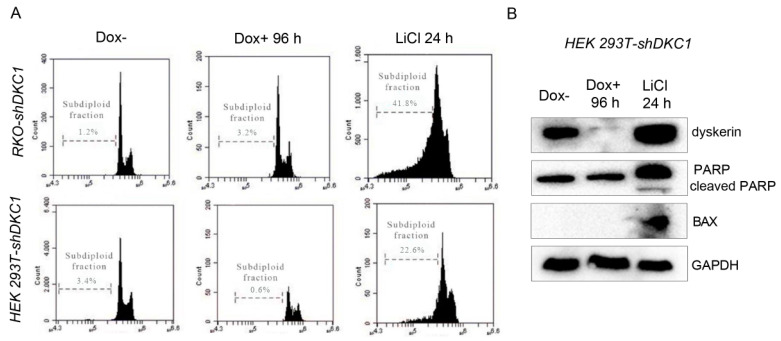
Dyskerin depletion does not activate apoptosis. (**A**,**B**) Induction of apoptosis was measured by calculating the percentage of control (Dox–) and silenced (Dox+) cells displaying subdiploid DNA content following propidium iodide staining. As a positive control, RKO-shDKC1 and HEK 293T-shDKC1 cells were treated for 24 h with LiCl (60 mM), an inducer of apoptosis. (**C**) Western blot analyses of the apoptotic markers PARP and BAX in control (Dox–), dyskerin-depleted cells (Dox+ 96 h), and control cells treated with LiCl (60 mM) for 24 h; the latter served as a positive control.

**Figure 4 biomedicines-10-01092-f004:**
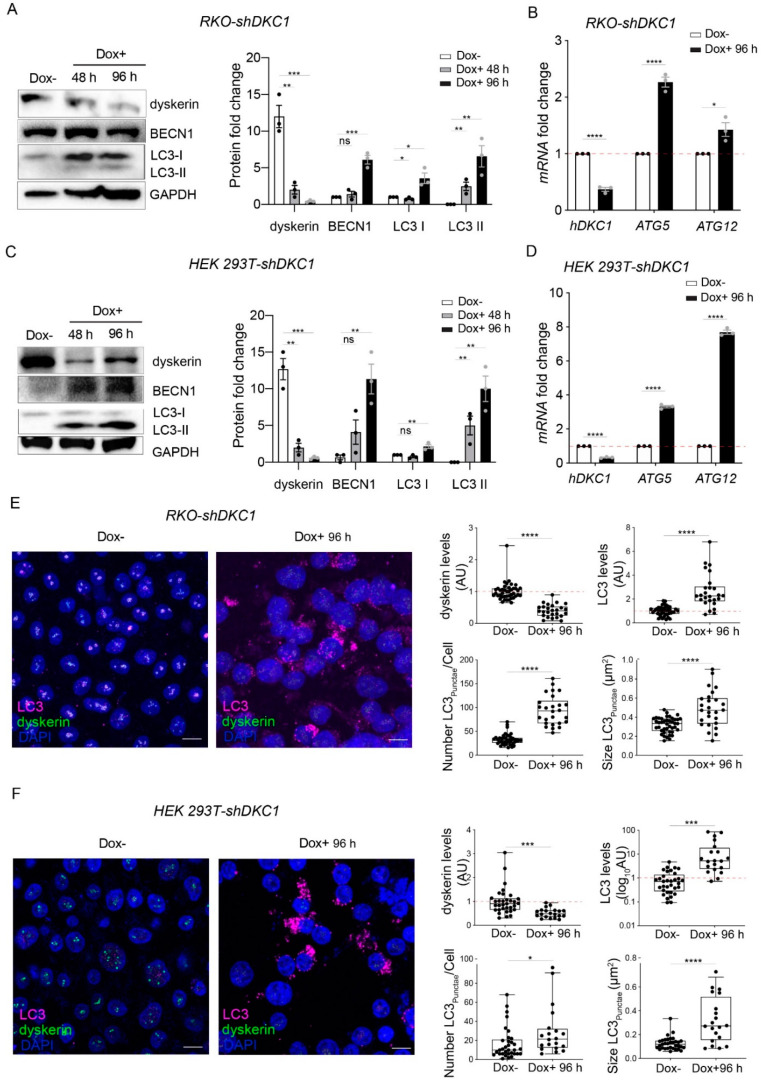
Dyskerin downregulation induces a set of autophagic markers. (**A**,**C**) Western blot analyses of BECN1 and LC3 proteins following 48 or 96 h of dyskerin depletion. Dyskerin depletion induces LC3-II in both RKO-shDKC1 (**A**) and HEK 293T-shDKC1 (**C**) silenced cells (Dox+). Upon silencing, LC3-I, whose level is not indicative of autophagic induction, also appeared upregulated in RKO cells. BECN1 levels increased in both cellular systems at the times considered. Protein levels were normalized to GAPDH. Results are expressed as mean ± SEM (*n* = 3). (**B**,**D**) qRT-PCR analyses in both RKO-shDKC1 and HEK 293T-shDKC1 ((**B**,**D**), respectively) cells show an increase in the transcription of ATG12 and ATG5 genes, which participate in the elongation of autophagosomes, following 96 h of Dox-induced dyskerin silencing. Results are expressed as mean ± SEM (*n* = 3). (**E**,**F**) Confocal micrographs of control (Dox–) and silenced (Dox+) RKO-shDKC1 (**E**) and HEK 293T-shDKC1 (**F**) cells, immunolabeled with anti-LC3 (magenta) and anti-dyskerin (green) antibodies. Scale bars 10 μm. Nuclei were counterstained with DAPI. LC3 and dyskerin levels were quantified in multiple cells (*n* = 42 RKO-shDKC1 Dox–; *n* = 26 RKO-shDKC1 Dox+ 96 h; *n* = 33 HEK 293T-shDKC1 Dox–; *n* = 20 293T HEK 293T-shDKC1 Dox+ 96 h), normalized and reported relative to the Dox– condition as mean ± SEM. The cytoplasmic LC3 puncta number and size were also calculated for the same cells. Both LC3 puncta and size were significantly upregulated after 96 h of Dox treatment. An unpaired one-tailed *t*-test was used in pairwise comparisons with respect to the Dox– condition in all statistical analyses in this figure (**** *p* < 0.0001; *** *p* < 0.001; ** *p* < 0.01; * *p* < 0.05; ns, *p* > 0.05).

**Figure 5 biomedicines-10-01092-f005:**
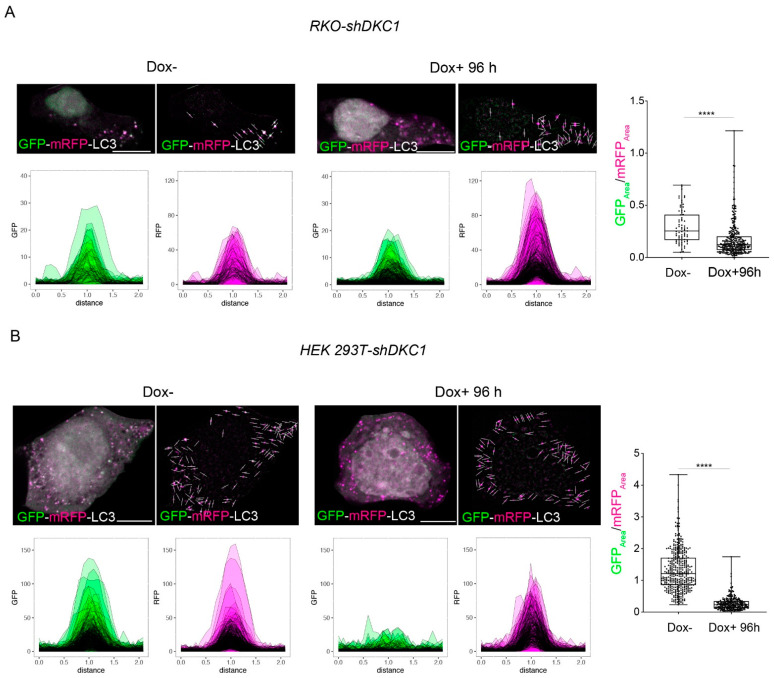
Dyskerin depletion promotes a functional autophagic flux. RKO-shDKC1 and HEK 293 T-shDKC1 Dox– cells were transiently transfected with the GFP-mRFP-LC3 tandem construct (containing LC3-I fused to mRFP and GFP). After 5 h of transient transfection, untreated (Dox–/GFP-mRFP-LC3) or treated (Dox+/GFP-mRFP-LC3) cells were grown for 96 h and then fixed and analyzed by confocal microscopy to detect individual GFP-mRFP-LC3 puncta (see materials and methods). For each single punctum, the area under the plot-profile in both GFP and mRFP channels and the GFPArea/RFPArea ratios were calculated. (**A**) GFP-mRFP-LC3 puncta analyses in RKO-sh-DKC1 cells. One representative cell is shown per condition (left panels) and linear ROIs centred on each of detected punctum in this cell are shown as well (right panel). Scale bars 10 μm. GFP and mRFP plot-profile curves for all the detected puncta are shown for both conditions (*n* = 73 puncta from 5 RKO-shDKC1 Dox– cells; *n*= 316 puncta from 5 RKO-shDKC1 Dox+ 96 h cells). The GFPArea/RFPArea ratio is significantly decreased after 96 h of Dox treatment, demonstrating the majority of the LC3 puncta detected in DKC1 silenced cells represent phagosomes fused with lysosomes. (**B**) GFP-mRFP-LC3 puncta analyses in HEK 293T-shDKC1 cells. Scale bars 10 μm. The GFPArea/RFPArea ratio is significantly reduced after 96 h of Dox treatment, demonstrating an increased autophagic flux upon dyskerin depletion (*n* = 400 puncta from 5 HEK 293T-shDKC1 Dox– cells; *n* = 264 puncta from 5 HEK 293T-shDKC1 Dox+ 96 h cells). All statistical analyses were performed using an unpaired one-tailed *t*-test in pairwise comparisons with respect to the Dox– condition (**** *p* < 0.0001).

**Figure 6 biomedicines-10-01092-f006:**
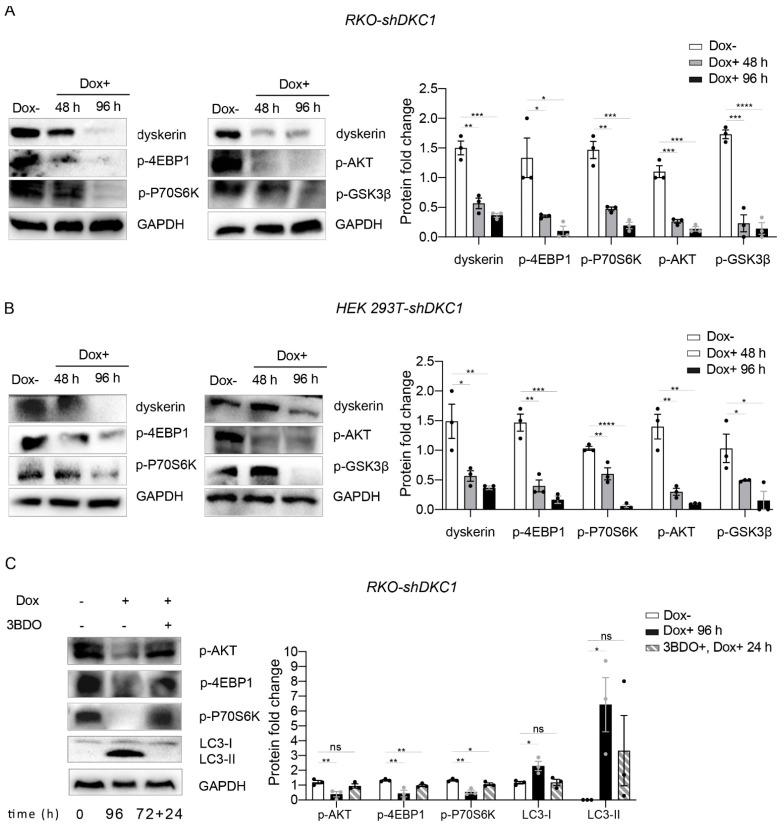
Dyskerin downregulation inhibits AKT/mTOR signaling. (**A**,**B**) Western blots showing the downregulation of p-4EBP1, p-P70S6K, p-AKT and p-GSK3β upon dyskerin depletion. Band intensities for each protein were normalized to the band intensity of GAPDH and the results are expressed as mean ± SEM (*n* = 3). (**C**) The 3BDO treatment rescued the expression levels of p-AKT, p-4EBP1 and p-P70S6K and prevented the accumulation of LC3 II. In the timeline, 72 + 24 h indicates that 72 h of Dox treatment were followed by 24 h of Dox + 3BDO treatment. For quantification, the band intensity for each protein was normalized to the band intensity of GAPDH. Results are expressed as mean ± SEM (*n* = 3). An unpaired one-tailed *t*-test was used in pairwise comparisons with respect to the Dox– condition (**** *p* < 0.0001; *** *p* < 0.001; ** *p* < 0.01; * *p* < 0.05; ns, *p* > 0.05).

## Data Availability

The data presented in this study are available upon request from the corresponding author.

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
