# Peer review of "Dyskerin Downregulation Can Induce ER Stress and Promote Autophagy via AKT-mTOR Signaling Deregulation"

_biomedicines, 2022, doi:10.3390/biomedicines10051092_

Round 1
Reviewer 1 Report
The mature tetrameric H/ACA protein complex is made up of dyskerin, NOP10, Nhp2 and Gar1. Two sets of the tetrameric H/ACA proteins bind to a small RNA with the cognate hinge box-ACA sequence motif to form the functional ribonucleoprotein (RNP), and the functionality of the RNP complex is determined by the identity of the small RNAs in the complex. The H/ACA-RNPs serve two critical biological functions in the human soma. As part of the telomerase holoenzyme, H/ACA proteins are complexed with telomerase RNA (TER) and are essential for this telomerase subunit's stability and the catalytic activity of the assembled telomerase. H/ACA-RNP is also complex with other non-coding RNAs, including sno- and scaRNAs. With these ncRNAs pairing with their specific target RNAs (rRNAs and snRNAs, respectively), the catalytic component dyskerin performs base modifications of specific uridine residues to pseudouridine. Almost all X-linked Dyskeratosis Congenita (XDC) cases are caused by single amino acid substitutions in the dyskerin protein, located in two mutational hotspots in the N- and the C-terminal of the polypeptide. XDC mutations reduced cellular accumulation of TER, resulting in sub-optimal telomerase activity and compromised telomere maintenance. Mutant dyskerins universally affect TER accumulations and telomerase functions, with only a handful of studies reported XDC -associated sno/scaRNA stability or functional defects.
In this manuscript, the authors followed up on their previous work investigating the effects of inducible (with doxycycline) siRNA-KD on endogenous dyskerin in two human cell models: 293HEK cells and RKO colorectal cancer cells. Within 96h of dyskerin KD, the authors reported a time-dependent accumulation of the heat-shock proteins HSP70 and 90 and an accumulation of the ER-stress marker CRT (Figure 1), suggesting the immediate engagement of stress-response in dyskerin KD cells. Dyskerin-KD markedly induced the immunofluorescent signals of GRP78, a master regulator of the Unfolded Protein Response, particularly the PERK-related UPR pathway (Figure 2). Instead of apoptosis (Figure 3), dyskerin-KD induced the onset of autophagy, as measured with the increased protein accumulation of BECN1, mRNA expression of ATG5 and 12 and the immunofluorescent detection of LC3 foci (Figure 4); and a GFP-mRFP-LC3 detection assay confirmed the onset of autophagic flux, following dyskerin shRNA-induction (Figure 5). Finally, transient dyskerin-KD induced the silencing of the mTOR signalling (through down-regulation of p-AKT, p-4EBP1, p-P70S6K and p-GSK3b; signals of which could be rescued with an mTOR agonist 3BDO. Treatment with 3BDO partially reduced the autophagy signal as measured with LC3 accumulation and cleavage (Figure 7).
This is a very straightforward biochemical and cell biology study. The generation methods and validation of the inducible dyskerin shRNA expression models were previously reported (FEBS Open Biology 2017). As a basic science study evaluating the cellular response to transient depletion of dyskerin, I found the connection with the unfold protein response and autophagy shown here to be convincing. There are some problems with some western blot images, but the most critical issue is that I found the interpretation of the findings regarding their clinical relevance to X-DC pathologies to be misleading. I have summarized these issues below:
1) Almost all X-linked DC cases are caused by single amino acid substitutions in the dyskerin protein; these mutations do not affect steady-state protein levels. Out of the 40-plus known XDC mutations, only two mutations, one in the promoter region and the other a deletion mutant in Exon 15, are predicted to cause reduced dyskerin levels. However, these germline mutations are expected to behave very differently from an acute loss of dyskerin expression. There was an early debate about whether XDC is caused by telomerase deficiency or the loss of proper pseudouridine modification activities in the non-coding RNAs (rRNAs and snRNAs). With the discovery of other inheritance forms of DC, which are almost exclusively caused by mutations in genes essential to telomere maintenance, it is now generally accepted that DC is a telomere-maintenance disorder. The similarity in molecular features, and overlapping clinical presentations between the X-linked form and other inheritance forms of DC, provide little support to pseudouridine modification dysfunction as a unifying mechanism for all XDC cases.
Laboratory models of dyskerin knockout are not compatible with life in higher eukaryotes. In essence, while telomerase deficiency caused by dyskerin mutations with incomplete penetrance underlies XDC, the complete loss of dyskerin expression causes embryonic lethality and cannot be tolerated. Thus, this reviewer strongly believes that transient dyskerin depletion in transformed human cells cannot be an accurate model for XDC. Current literature reporting the study of XDC molecular phenotypes typically employed a knock-in+knockout scheme, where shRNA-mediated KD of wildtype dyskerin expression is replaced with the recombinant expression of XDC-dyskerin mutants (see MacNeil et al., NAR 2019). The authors should be wary of overstating a claim of direct clinical correlation. Doing so would be misleading and cause unnecessary confusion for the general audience. I strongly recommend that the language throughout the manuscript be extensively modified.
2) Loss of rRNA pseudouridine modifications had been previously implicated in translation dysfunctions, including reading-frame shift, loss of IRES-mediated translation and amino acids fidelity. Accordingly, the observations by the authors upon transient dyskerin-KD are completely in-line with what is previously known in the literature. The value of the current study lies in the connection between dyskerin-KD and the early response by engaging the autophagy pathway towards rescuing ER stress. While autophagy dominates early responses to ER stress associated with dyskerin-KD, prolonged UPR engagement will conceivably lead to loss of cell viability. This is particularly crucial, as in the cases of hypomorphic XDC allele, the loss in dyskerin expression is chronic. It will be interesting to extend the current time-course study, include late response to the loss of dyskerin expression, and compare the phenotypes between transformed and non-transformed cells upon prolonged loss of dyskerin expression.
3) This reviewer appreciates the inclusion of all western blot data for review. However, some of the western blot data are of poor quality, with multiple bands on the same blots (GRP78, ATF4, eIF2A, PERK, BECN1, p-P70S6K, AKT) and over-exposure of signals that made hard to imagine how accurate quantification could be conducted (GRP78, PERK, CHOP, BECN). The authors should improve on these technical issues.
Author Response
"Please see the attachment."

Reviewer 2 Report
The contents and scientific sounds are high. Overall, however, the manuscript is so long and include many unimportant sentences and overlapping. Your “Results” section also includes some discussions which should be mentioned in the “Discussion” section. So, you have to brush them up to publish. Moreover, there are many careless easy mistakes. You and co-authors must check again and again before submission.
Introduction:
L79, ER, here, should be explained as an abbreviation.
Materials and Methods:
L102, P/S should be explained as an abbreviation.
L104, You can add about the timing of exchange of medium with puromycin during incubation.
L215-217, 225-227, 248-250, Transfer to “Statistical Analysis” sub-section.
2.10. How many times did the experiments include for the statistical analysis? You can mention about the number of experiments you used for analysis.
Results:
3.1. You need to shorten paragraphs and more briefly summarize the essence what the result of the experiment indicated. For a part of example, L235-237 is not needed in here, because this explanation was already mentioned in M&M section as you wrote.
Figure 1; Unimportant data, like L303 and 304 (this is not the all), should not be described as the Figure legend. You also have to explain the abbreviation of LUT.
L322, Use CRT, not calreticulin.
3.2. Don’t use UPR here.
L335, The abbreviation of ER should be indicated and explained in the introduction section.
3.3. You should briefly explain the roll of LiCl here.
Figure 3(B), You have to add quantitative statistical analysis. Where is Figure 3(C)?
Figure 5, Which type of LC3 (type-I or II)? You can clarify the type.
Figure 6(A), The protein expression level of p-GSK3B (Dox+ 96h) looks higher than that of Dox+ 48h. This might be unclear. You should do the same experiment and analysis to confirm the expressions.
Figure 6(C), For the western membrane, what is the “time” you described in the Figure? What treatment time? It’s unclear. You must clearly explain the experimental planning in the M&M section or in the legend.
Discussion:
L621, Regarding Akt/mTOR pathway, Okuyama, et al. also reported the cetuximab, an EGFR inhibotor-dependent inhibition of cell migration, resulting in high cell density-related cell stress and persistent cell-cycle arrest at G1 phase culminating in autophagy.
They have similar points of view. You can add their contents and combining the result of your study, you can additionally discuss about the effect to cell migration and cell cycle.
Okuyama K, Suzuki K, Naruse T, Tsuchihashi H, Yanamoto S, Kaida A, Miura M, Umeda M, Yamashita S. Prolonged cetuximab treatment promotes p27 Kip1-mediated G1 arrest and autophagy in head and neck squamous cell carcinoma. Sci Rep. 2021, 4; 11(1): 5259.
Author Response
"Please see the attachment."

Round 2
Reviewer 1 Report
I appreciate the revised content, and the authors' more measured discussions on their current findings' on XDC disease etiology.
Reviewer 2 Report
I think the manuscript is sufficiently improved and is worth publishing in the journal.
Just one point, L261, "Immunofluorescent staining" is better. This change may be able to be corrected at the proof making.